# Spatial distribution of health-risky road traffic noise pollution in Dessie City, North East Ethiopia

Getahun Gebre Bogale[1]*, Tadesse Sisay[2], Asnakew Molla Mekonen[3], Muluken Tessema Aemiro[4]

1 Department of Health Informatics, School of Public Health, College of Medicine and Health Sciences, Wollo University, Dessie, Ethiopia, 2 Department of Environmental Health, College of Medicine and Health Sciences, Wollo University, Dessie, Ethiopia, 3 Department of Health Systems Management, School of Public Health, College of Medicine and Health Sciences, Wollo University, Dessie, Ethiopia, 4 Department of Public Health, College of Health Sciences, Institute of Medicine and Health Sciences, Debre Berhan University, Debre Berhan, Ethiopia

* getahungebre21@gmail.com

**Data Availability Statement:** All relevant data are within the paper and its Supporting Information files.

**Funding:** The author(s) received no specific funding for this work.

## Abstract

### Objective

Dessie is the trade center for northeast Ethiopia. High traffic flow plus overacting of promotion made the city noisy. There is a shortage of relevant evidence that enforces policy makers to design intervention plans. Therefore, this study aimed to explore the health-risky road traffic noise pollution in Dessie City, Ethiopia.

### Methods

The study was conducted by purposive selection of the study area and sampling sites of the city from May 31, 2021 –June 6, 2021. Noise level recordings were taken by a digital Sound Meter and location data was collected by Global Positioning System. Residential, health facility, commercial, and mixed sites were identified by field observation. A total of 20 noise sampling points were included. The sampling points were selected by considering World Health Organization guideline. The measurements were taken twice a day at peak hours, between 8:00–11:00am and 4:00–7:00pm on all days of the week. The sound level meter was placed at a height of 1.5m and 2m from the curb. A total of 280 sound level records were conducted over one week.

### Results

Among twenty noise recording sites, more than 50% of them registered as excessive noisy sites for all types of site categories (health facility, residential, commercial, and mixed areas). For the seven days, average noise recordings were in the range of 66–72 dB at 83% of mixed areas; 33% of health facilities; 25% of residential areas, and 86% of commercial areas. The highest levels of noise pollution were seen at the Bus-station, *Buanbuawuha* Square, *Tekuam*, *Arada*, Ethio General Hospital, *Ersha-seble*, and *Menafesha* areas.

**Competing interests:** The author has declared that no competing interests exist.

**Abbreviations:** dB, decibel; GPS, Global Positioning System; L10, A-weighted ten-percentile noise level; LAeq, A-weighted equivalent continuous noise level; Lmax, Instantaneous maximum noise level; WHO, World Health Organization.

## Conclusion

This study shows that the average noise level measurement within a week exceeded the permissible limits set by Ethiopia and the World Health Organization. It helps for policy development and timely actions against noise pollution and as baseline information for further investigation.

## 1. Introduction

Noise is unwanted sound and a serious cause of global worry, especially in urban areas of developing and developed nations [1]. Noise pollution is a significant environmental problem in many rapidly urbanizing areas. Environmental noise pollution, a form of air pollution, is a threat to health and wellbeing. It is more severe and widespread than ever before, and it will continue to increase in magnitude and severity because of population growth, urbanization, and the associated growth in the use of increasingly powerful, varied, and highly mobile sources of noise [2]. This problem is not recognized despite the fact that it is steadily growing in developing countries [3].

According to Robert Koch "*A day will come, man will have to fight merciless noise as the worst enemy of health*". The major cities of the world are now facing the problem of the rise in noise pollution due to very high population, transportation, congestion and associated commercial and industrial activities. Though, the urban population is much more affected by such pollution; however, small town/villages along side roads or industries are also a victim of this problem [4]. Excessive noise is a global occupational health hazard with considerable social and physiological impacts, including noise-induced hearing loss [5], cardiovascular diseases, cognitive impairment in children, sleep disturbance, tinnitus, and annoyance among the exposed groups [6]. The major sources of noise are vehicles, musical instruments, small scale industries, bars, night clubs, religious speakers, urbanization, and human activities [4, 7]. Among urban noise sources, road traffic noise is the highest contributor to noise pollution. It is also a big challenge for urban planners and environmental engineers to overcome road traffic noise in cities [8].

Tens of millions of Americans suffer from a range of adverse health outcomes due to noise exposure, including heart disease and hearing loss [9]. According to a European Union (EU) publication, about 40% of the population in EU countries is exposed to road traffic noise at levels exceeding 55 dB; 20% is exposed to levels exceeding 65 dB during the daytime; and more than 30% is exposed to levels exceeding 55 dB at night [10]. Noise pollution is the least addressed issue in Ethiopia and in Africa in general [11].

In the developed world, a lot of actions such as noise pollution control legislations, regulations, and noise policies have been taken to minimize the problems of noise pollution [12]. Among developing countries, Ethiopia has established a comprehensive environmental policy in which the overall policy goal is aimed at improving and enhancing the health and quality of life of people. One of the objectives of Environmental Pollution Control Proclamation 300/2002 is to control noise pollution [13]. Though there have been a policy and laws addressed to noise pollution which have never been implemented properly due to the lack of programs, this country has not yet fully recognized noise pollution as human health risk factors [13–17]. The amount of noise in Addis Ababa from such sources is the day-to-day grumbles of the residents [18, 19]. The report presented by Mahlet G., for the Forum for Environment confirmed that noise pollution in Addis exceeded the standard given by WHO (1999) [20].

A study done in Dire Dawa City indicated that the average noise level measured at commercial, residential and mixed sites was higher than the acceptable limit set by WHO [21]. The trend is almost the same in the studied metropolitan cities of Ethiopia. Since Dessie is the trade center for North East Ethiopia, high traffic flow plus overacting of promotion made the city noisy. Beyond the above-mentioned little evidence, there are no studies conducted in this city which show the issue and force policy makers to take measures. Thus, studying this issue spatially is essential for both timely actions against noise pollution and as baseline information for further investigation.

## 2. Methods and materials

### 2.1. Study setting, period, and materials

Dessie City was founded in 1882. Dessie is located in the northern part of Ethiopia in the Amhara National Regional State, South Wollo Zone, at a distance of 400 km from Addis Ababa. Its astronomical location is 11˚08' North Latitude and 39˚38' East Longitude. Dessie is one of the reform towns in the region and has a city administration consisting of a municipality and urban and rural *Kebeles* (*the lowest administrative level in Ethiopia*). The city has a structural plan which was prepared in 2010 [22]. According to Central Statistics Agency 2014 E.C projection, the total population of the city is 285,530 (Dessie City Administration Health Department 2014 fiscal year plan, unpublished); it has four gates of roads from different directions. There are more than four main roads within the city. The study was conducted by purposive selection of the study area and sampling sites of the city from May 31, 2021 –June 6, 2021. Both Dessie city polygon and its main road polylines were obtained by digitization from Google Earth Pro.

### 2.2. Sample size determination

By considering the main roads, residential, commercial, and mixed sites were identified by field observation. A total of 20 sampling points were selected for measuring sound levels using a calibrated scientific digital sound level meter. The sampling points where noise pollution was expected were selected by considering World Health Organization guideline and professional judgments (which means that the author decided to select some sampling points as per the knowledge/experience of the author and current situations of the field sites). The measurements were taken twice a day at peak hours, between 8:00–11:00am and 4:00–7:00pm using a digital Sper Scientific Sound Meter 840029 on all days of the week. The sampling time for each measurement was 30 minutes. The sound level meter was placed at a height of 1.5m and 2m from the curb. So, a total of 280 sound level records were conducted over one week. The global Positioning System (GPS) was used to collect coordinates' data from twenty noise level measurement points.

A total of six recording sites (#1, 2, 5, 8, 12, and 15) are mixed (both residential and commercial) areas. Three recording sites (#9, 10, and 11) are areas of health facilities having loads of patients flow. Four recording sites (#13, 17, 18, and 20) are residential areas. Seven recording sites (#3, 4, 6, 7, 14, 16, and 19) are commercial areas.

### 2.3. Operational definitions

- A day's average noise level recording means the average of Morning (8:00am– 11:00am) and Afternoon (4:00pm– 7:00pm) recordings per sampling site.

- A week's average noise level recording is the summation of average recordings of each day per week.

## 2.4. Data collection procedures, tools and quality control

Data collectors and supervisors were trained on the objective of the study, how to take noise levels, and coordinate data using a noise level meter and GPS, respectively. The noise level meter and GPS tools were checked for calibrations. Noise level recordings were taken with calibrated digital Sper Scientific Sound Meter 840029 and location data were collected by BHCnav pro F78 Global Positioning System. Noise level and location data were recorded both in digital and respective paper-based recording formats. The investigator and supervisors supervised the data collection and took immediate corrections upon errors were introduced.

## 2.5. Data analysis and interpretation

Since road traffic noise is continuing sound, the noise parameters and statistical parameters were measured and described in terms of noise level standard values such as A-weighted equivalent continuous noise level ($L_{Aeq}$), A-weighted ten-percentile noise level ($L_{10}$), and instantaneous maximum noise level ($L_{max}$) [23].

A day's average noise levels were calculated by Eq (1) and a week's average noise levels were computed by Eq (2) below.

$$\text{A day average noise level recording (dB) for } Day_i = \frac{M_r + A_r}{2} \qquad \text{Eq(1)}$$

$$\text{A week average noise level recording (dB)} = \frac{\sum Day_i}{7 \ days} \qquad \text{Eq(2)}$$

Where,

○ $Day_i$ represents day 1, 2, 3, . . ., 7

○ $M_r$ is Morning record for each day

○ $A_r$ Afternoon record for each day

The noise levels were logarithmically averaged (L) as L1, L2, L3,——Ln.

$$\text{Average noise level (L)} = L_{eq} = 10 Log \left[ \frac{\sum_{i=1}^{n} 10^{Leqi/10}}{N} \right], \qquad \text{Eq(3)}$$

Where,
L = Average noise level (dBA) or Leq = Equivalent noise levels,
L1, L2 . . .Ln = Observed noise levels from 1 to n[th], (in dBA).
N = Total number of observed noise levels.

During all processes of calculation, the order of the recorded noise values was maintained properly. The measurement was taken in calm and non-disturbing conditions, i.e. in the absence of rain, wind disturbance and other interfering noise generating activities nearby.

## 2.6. Ethics approval and consent to participate

Ethical clearance was obtained from the Ethical Review Committee (**Ref.No CMHS 537/13/13**) of College of Medicine and Health Sciences, Wollo University. Support letter was obtained from Dessie City Administration Health Department.

## 3. Results

Among twenty noise recording sites (named as Sites 1 and 2: Arada; Site 3: Gimruk; Site 4: Salayish; Site 5: Piassa-Taxi queue to Tequam; Site 6: Piassa to bus station road; Site 7: Gate of bus station; Site 8: Fasika Hotel; Site 9: Gate of Selam General Hospital; Site 10: Gate of Dessie Comprehensive Specialized Hospital; Site 11: Gate of Ethio General Hospital; Site 12: Abebech Wollo–Menafesha to St. Gebriel road; Site 13: In front of St. Gebriel Church; Site 14: Buanbuwa wuha Square–in front of Dashen Bank district office; Site 15: Buanbuwa wuha Condominium; Site 16: Dolphin–United Bank-Buanbuawuha branch; Site 17: Gate of Red Cross Condominium; Site 18: Terminal Condominium–Tekuam to Piassa Road; Site 19: Tekuam–in front of Abyssinia Bank; and Site 20: Ersha-Seble Condominium), more than 50% of them registered as excessive noisy sites for all types of site categories (health facility, residential, commercial, and mixed areas). Noise recording was taken for seven days from Monday to Sunday two times a day and average values were taken for this report. The day time (morning and afternoon) noise levels were taken in terms of noise statistical parameters such as $L_{Aeq}$ $L_{10}$, and $L_{max}$ (**Fig 1, S1 Table**).

On Monday, in general, noise levels were in the range of 66–73 decibels (dB) at study sites 1–9, 11, 13, 14, and 19. Noise levels were in the range of 62–65 decibels at study sites 10, 15–18, and 20. Specifically, noise levels were above 65 dB at four (67%) of mixed areas; six (86%) of commercial areas; one (25%) of residential areas; and two (67%) of health facility areas. Noise levels were above 55 dB at one (33%) of sampled health facility areas; two (33%) of mixed areas; one (14%) of commercial areas; and three (75%) of residential areas On Tuesday, noise

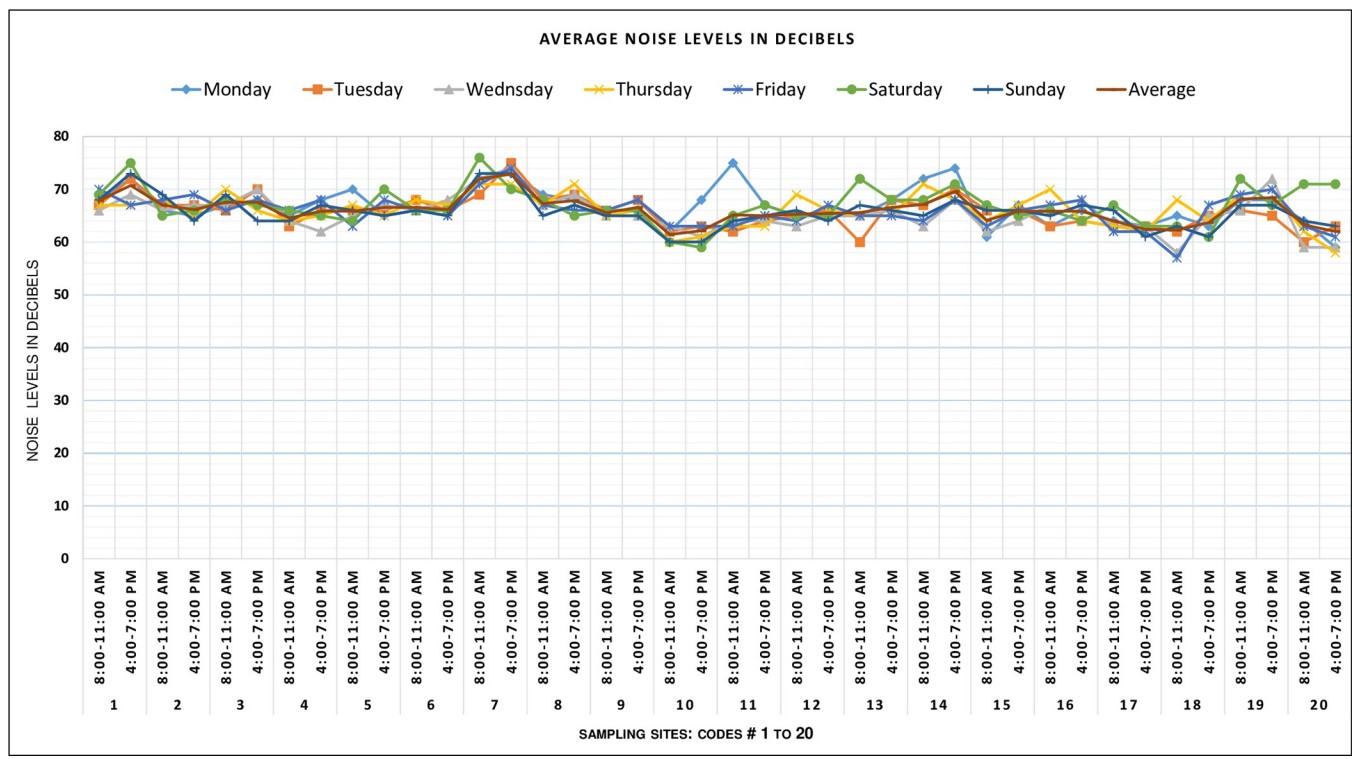

**Fig 1. Daily average (morning and afternoon) and a week average road traffic noise levels in decibels in Dessie city administration, northeast Ethiopia.** (Sampling sites 1: Arada 1; 2: Arada 2; 3: Gumruk; 4: Salayish; 5: Piassa; 6: Piassa to bus station; 7: Bus station; 8: Fasika Hotel; 9: Selam G. Hospital; 10: Dessie Comprehensive Specialized Hospital; 11: Ethio G. Hospital; 12: Abebech Wollo; 13: St. Gebriel Church; 14: B/Wuha Square; 15: B/Wuha Condominium; 16: Dolphin; 17: Red Cross Condominium; 18: Terminal Condominium; 19: Tequam; 20: Ersha-seble Condominium).

levels registered from 66–72 dB at all mixed areas; one (33%) of health facilities areas; and five (71%) of commercial areas. Noise levels ranged from 62–65 dB at the rest of the study sites (particularly hospitals, mixed, and residential areas) (**Fig 1**, **S1 Table**).

On Wednesday, noise level readings were in the range of 66–73 dB at all mixed areas; one (25%) of residential areas; and six (86%) of commercial areas. The rest of the areas registered from 59–65 dB noise levels. On Thursday, noise levels registered from 66–72 dB at all mixed areas; one (33%) of health facilities areas; one (25%) residential areas and six (86%) of commercial areas. Noise levels ranged from 60–65 dB at the rest of the study sites (particularly hospitals, mixed, and residential areas) (**Fig 1**, **S1 Table**).

On Friday, noise levels were from 66–73 dB at all mixed areas; one (33%) of health facilities areas; and at all commercial areas. Noise levels ranged from 62–65 dB at the rest of the study sites (particularly hospitals, mixed, and residential areas). On Saturday, noise levels were from 66–73 dB at all mixed areas; two (67%) of health facilities areas; two (50%) of residential areas, and six (86%) of commercial areas. Noise levels ranged from 60–65 dB at the rest of the study sites (particularly hospitals, mixed, and residential areas) (**Fig 1**, **S1 Table**).

On Sunday, noise levels were recorded from 66–73 dB at all sampled mixed areas; one (25%) of residential areas; and at all commercial areas. The rest recordings ranged from 60–65 dB at all sampled health facilities and the rest of the residential areas. Over all, seven days (one week), average noise recordings were in the range of 66–72 dB at five (83%) of mixed areas; one (33%) of health facilities; one (25%) of residential areas; and six (86%) of commercial areas (**Fig 1**, **Table 1**).

## 4. Discussion

This study was conducted at residential, health facility, commercial, and mixed sites purposively selected from 20 areas in the city administration of Dessie, one of the largest cities in the Amhara region, Ethiopia. As indicated in **Table 1**, the all over noise measurements in the study area exceeded the national and international permissible limits set by Ethiopia for residential (55 dB) and commercial (65 dB) [24], and World Health Organization (45 dB and 70 dB) [23]. It is in line with studies done in Addis Abeba [19, 20] and Dire Dawa City Administrations [21, 25, 26]. It may be due to the fact that the city is surrounded by mountains that may aggravate noise pollution [27]; landscape imbalance and proliferation of old cars may force it to emit noisy sounds. Moreover, close proximity of residences to the roadway in mountainous areas exposes the residents to higher levels of traffic noise [27].

Over all, seven days (one week), average noise recordings were in the range of 66–72 dB at five (83%) of mixed areas; one (33%) of health facilities; one (25%) of residential areas; and six (86%) of commercial areas. The results of this study tell us that the sampling areas are noise zones which resemble the definitions given by the International Organization for Standardization (ISO 1996–1,2) [28, 29]. The measured average noise level value for the residential site is in the range of 63–66 dB; the highest level being in front of St. Gebriel Church. For mixed

**Table 1. A week average measurement of noise pollution levels (dB) in the four sites of Dessie city, northeast Ethiopia, in comparison to the Ethiopian and WHO standards [24].**

| Categories of sites | Measured Noise Levels (dB) | | Ethiopian Standard (dB) | WHO Standard (dB) |
|---|---|---|---|---|
| | Maximum | Minimum | | |
| Residential | 66 | 63 | 55 | 45 |
| Health facility | 66 | 62 | = | = |
| Commercial | 72 | 65 | 65 | 70 |
| Mixed | 69 | 65 | = | = |

sites, it was in the range of 66–69 dB, the highest-rated around the Arada terminal, which is one of the hot spots for business activities. For commercial sites, the noise reading was in the range of 65–72 dB; the highest noise level was around the bus station, which is the main traffic flow and business running area. Regarding health facility sites, noise measurements ranged from 62–66 dB; the highest noise level was recorded at Selam General Hospital followed by Ethio General Hospital.

When comparing recordings among the seven days, except Tuesday and Thursday, all the other days of the week had the highest levels of readings (70–73 dB); seen at the bus station, Buanbuawuha Square, Tekuam, Arada, Ethio General Hospital, Ersha-seble, and Menafesha sites. This may be due to the fact that these days are normal trading days in the locality; people gathering from the city surroundings and urban areas and vehicles can aggravate the noise pollution. Even though Sunday's high noise pollution is unexpected, many vehicles are coming in and going out to/from the bus station as usual, which may have contributed to the pollution.

In general, this study shows that residents exposed to excessive noise in residential and mixed areas where noise levels are above the limits. The results are even above the noise standard values of $L_{Aeq}$. which are 65 dB and 70 dB for urban residential and commercial and mixed areas at day time, respectively [23, 30]. Thus, they need great attention through developing preventive and mitigation policies and guidelines. The community, due to the lack of awareness and training towards the risks of noise pollution, do not take any safety precautions and are vulnerable to irreversible harmful impacts. Hence, proper urban planning, formulating implementable and acceptable laws and standards, and providing community awareness through training are needed to protect the public against noise pollution which should be maintained at the level of the permissible limits. Moreover, the author recommends the affected people to use earplugs for high-risk areas; shut the door at the time of high traffic load; stay away from noisy areas; and the administration to prohibit car horn, control and follow noise levels, and planting trees near sensitive areas.

## 5. Conclusions

This study shows that the average noise level measurement within a week exceeded the permissible limits set by Ethiopia and the World Health Organization. The highest levels of noise pollution were seen at the bus station, Buanbuawuha Square, Tekuam, Arada, Ethio General Hospital, Ersha-seble, and Menafesha areas. By nature, noise cannot be diluted, cleansed, collected or reused, but a precautionary principle and maximization approach can be applied, so that no one should involuntarily be exposed to excessive noise that could be harmful to his/her hearing, health, and wellbeing. The city administration shall design mitigation procedures and implement it to reduce the noise pollution in residential, health facility, commercial, and mixed areas.

## Supporting information

**S1 Table. Details of noise level recordings from twenty sampling sites of Dessie city, northeast Ethiopia, 2021.** The table shows the maximum, the average and the minimum noise level measurements (in dB) for each sampling points across the seven days of the week.
(XLSX)

## Acknowledgments

We are very grateful to Wollo University for the approval of the ethical clearance. I would also like to thank my little brother Kidanewold Gebre and all other individuals participated in this study for their cooperation.

## Author Contributions

**Conceptualization:** Getahun Gebre Bogale, Tadesse Sisay.

**Data curation:** Getahun Gebre Bogale, Tadesse Sisay, Asnakew Molla Mekonen, Muluken Tessema Aemiro.

**Formal analysis:** Getahun Gebre Bogale, Tadesse Sisay, Asnakew Molla Mekonen, Muluken Tessema Aemiro.

**Funding acquisition:** Getahun Gebre Bogale.

**Investigation:** Getahun Gebre Bogale.

**Methodology:** Getahun Gebre Bogale, Tadesse Sisay, Asnakew Molla Mekonen, Muluken Tessema Aemiro.

**Project administration:** Getahun Gebre Bogale.

**Resources:** Getahun Gebre Bogale, Tadesse Sisay, Asnakew Molla Mekonen, Muluken Tessema Aemiro.

**Software:** Getahun Gebre Bogale.

**Supervision:** Getahun Gebre Bogale, Tadesse Sisay, Asnakew Molla Mekonen, Muluken Tessema Aemiro.

**Validation:** Getahun Gebre Bogale, Tadesse Sisay, Asnakew Molla Mekonen, Muluken Tessema Aemiro.

**Visualization:** Getahun Gebre Bogale, Asnakew Molla Mekonen, Muluken Tessema Aemiro.

**Writing – original draft:** Getahun Gebre Bogale, Tadesse Sisay.

**Writing – review & editing:** Getahun Gebre Bogale, Tadesse Sisay, Asnakew Molla Mekonen, Muluken Tessema Aemiro.

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
