## [Decision Letter · Decision Letter 0]

9 Dec 2021

PONE-D-21-28974Spatial distribution of health-risky road traffic noise pollution in Dessie City, North East EthiopiaPLOS ONE

Dear Dr. Bogale,

Thank you for submitting your manuscript to PLOS ONE. After careful consideration, we feel that it has merit but does not fully meet PLOS ONE’s publication criteria as it currently stands. Therefore, we invite you to submit a revised version of the manuscript that addresses the points raised during the review process.

Please see below the comments and suggested MAJOR revisions made by the individual(s) who reviewed your manuscript.  If provided, the referee's report(s) indicate the revisions that need to be made before it can be accepted for publication.

We look forward to receiving your revised manuscript.

Kind regards,

Ricardo Santos

Academic Editor

PLOS ONE

Journal Requirements:

4. We note that Figure 1 in your submission contain copyrighted images. All PLOS content is published under the Creative Commons Attribution License (CC BY 4.0), which means that the manuscript, images, and Supporting Information files will be freely available online, and any third party is permitted to access, download, copy, distribute, and use these materials in any way, even commercially, with proper attribution. For more information, see our copyright guidelines: http://journals.plos.org/plosone/s/licenses-and-copyright.

5. We note that Figures 2, 3, 4 & 5 in your submission contain [map/satellite] images which may be copyrighted. All PLOS content is published under the Creative Commons Attribution License (CC BY 4.0), which means that the manuscript, images, and Supporting Information files will be freely available online, and any third party is permitted to access, download, copy, distribute, and use these materials in any way, even commercially, with proper attribution. For these reasons, we cannot publish previously copyrighted maps or satellite images created using proprietary data, such as Google software (Google Maps, Street View, and Earth). For more information, see our copyright guidelines: http://journals.plos.org/plosone/s/licenses-and-copyright.

a. You may seek permission from the original copyright holder of Figure 2, 3, 4 & 5 to publish the content specifically under the CC BY 4.0 license.  

6.  We suggest you thoroughly copyedit your manuscript for language usage, spelling, and grammar. If you do not know anyone who can help you do this, you may wish to consider employing a professional scientific editing service. 

Whilst you may use any professional scientific editing service of your choice, PLOS has partnered with both American Journal Experts (AJE) and Editage to provide discounted services to PLOS authors. Both organizations have experience helping authors meet PLOS guidelines and can provide language editing, translation, manuscript formatting, and figure formatting to ensure your manuscript meets our submission guidelines. To take advantage of our partnership with AJE, visit the AJE website (http://aje.com/go/plos) for a 15% discount off AJE services. To take advantage of our partnership with Editage, visit the Editage website (www.editage.com) and enter referral code PLOSEDIT for a 15% discount off Editage services.  If the PLOS editorial team finds any language issues in text that either AJE or Editage has edited, the service provider will re-edit the text for free.

Reviewers' comments:

Reviewer's Responses to Questions

**Comments to the Author**

1. Is the manuscript technically sound, and do the data support the conclusions?

Reviewer #1: Partly

Reviewer #2: Partly

2. Has the statistical analysis been performed appropriately and rigorously? 

Reviewer #1: No

Reviewer #2: No

3. Have the authors made all data underlying the findings in their manuscript fully available?

Reviewer #1: No

Reviewer #2: No

4. Is the manuscript presented in an intelligible fashion and written in standard English?

Reviewer #1: Yes

Reviewer #2: No

5. Review Comments to the Author

Reviewer #1: The Authors are trying to gauge the road traffic noise pollution by evaluating the measured noise levels against the Ethiopian Standard and WHO Standard. However, the parameters of the criteria have not been indicated clearly. For instance, how do Ethiopian Standard and WHO Standard assess the noise levels? Is that Leq, L10, or L90? How long is the duration of the measurements? Noise is fluctuating energy, and thus the measurement duration plays an essential role for those statistical parameters. If the statistical parameter used to express the sound levels have not been indicated clearly, the measurement results are meaningless. Also, I suggest showing the measured noise levels tabulated in Table 1 to be illustrated in chart format for ease of reference.

Reviewer #2: 1) Overall, this paper presented the study on noise pollution in Dessie City by considering noise measurement at health facilities area, residential areas and commercial areas.

2) However, I did not find any contribution or new ideas presented in this paper. It is mainly reporting the outcomes of the measurement result without providing any new ideas or mitigation procedure to control the noise pollution.

3) The measuremen result does not compared with the established standard (ISO 1996-1, etc.). The authors should refers these established standard in the discussion.

4) The presented measurement result should also describe the noise in term of Lmax, L10, and LAeq as these are the standard value used in any noise analysis.

4) It is always good if the authors can provide a new idea or new mitigation procedure in order to reduce noise pollution in urban areas.

5) The authors mentioned about "professional judgement". I think the authors should described in details what type of judgement were considered here.

6) In my opinion, this paper required a lots of improvement in term of technical discussion, standard comparison and contribution in order to meet the requirement to publish in this Journal.

6. PLOS authors have the option to publish the peer review history of their article (what does this mean?). If published, this will include your full peer review and any attached files.

Reviewer #1: No

Reviewer #2: No

---

## [Author Response · Author response to Decision Letter 0]

21 Feb 2022

Dear Ricardo Santos,

I appreciate your decision to my manuscript and accept my apologies for the late response though I requested you to allow me further due dates due to my field trip. I also thank you all the reviewers for their valuable comments. 

Author response to Editor and Reviewers (note:- Comments=C1-20, Response=R1-20)

Manuscript ID: PONE-D-21-28974

Manuscript Title: Spatial distribution of health-risky road traffic noise pollution in Dessie City, North East Ethiopia

Author: Getahun Gebre Bogale 

Editor’s comments /queries 

C1. A rebuttal letter that responds to each point raised by the academic editor and reviewer(s). You should upload this letter as a separate file labeled 'Response to Reviewers'.

R1. Uploaded 

C2. A marked-up copy of your manuscript that highlights changes made to the original version. You should upload this as a separate file labeled 'Revised Manuscript with Track Changes'.

R2. It is done 

C3. An unmarked version of your revised paper without tracked changes. You should upload this as a separate file labeled 'Manuscript'. 

R3. Uploaded as “Manuscript”

Journal Requirements:

C4. Please ensure that your manuscript meets PLOS ONE's style requirements, including those for file naming. The PLOS ONE style templates can be found at 

R4. I confirm you that the manuscript is prepared as per the PLOS One template

C5. Please provide additional details regarding participant consent. In the ethics statement in the Methods and online submission information, please ensure that you have specified (1) whether consent was informed and (2) what type you obtained (for instance, written or verbal, and if verbal, how it was documented and witnessed). If your study included minors, state whether you obtained consent from parents or guardians. If the need for consent was waived by the ethics committee, please include this information.

R5. Thank you. Of course participants’ consent is mandatory if data are collected from human study participants. In the case of this study, only noise level data were measured/collected (using standardized noise level meter) from the hypothetical risky sites, not from human participants. However, the study was ethically approved by Ethical Review Committee of Wollo University and written permission was gained from Dessie City Administration Health Department. 

C6. If you are reporting a retrospective study of medical records or archived samples, please ensure that you have discussed whether all data were fully anonymized before you accessed them and/or whether the IRB or ethics committee waived the requirement for informed consent. If patients provided informed written consent to have data from their medical records used in research, please include this information.

R6. My study is not from a retrospective medical records or archived samples nor it is from patients. See the above response (R5).

C7. Your ethics statement should only appear in the Methods section of your manuscript. If your ethics statement is written in any section besides the Methods, please move it to the Methods section and delete it from any other section. Please ensure that your ethics statement is included in your manuscript, as the ethics statement entered into the online submission form will not be published alongside your manuscript. 

R7. I confirm you that the ethics statement is only appear in the methods section. 

C8. We note that Figure 1 in your submission contain copyrighted images. All PLOS content is published under the Creative Commons Attribution License (CC BY 4.0), which means that the manuscript, images, and Supporting Information files will be freely available online, and any third party is permitted to access, download, copy, distribute, and use these materials in any way, even commercially, with proper attribution. For more information, see our copyright guidelines: http://journals.plos.org/plosone/s/licenses-and-copyright . We require you to either (1) present written permission from the copyright holder to publish these figures specifically under the CC BY 4.0 license, or (2) remove the figures from your submission:

R8. Since the image is fitted from different images, it’s not possible to get permission from one copyright holder. So, I decided to remove from submission. In its place, I prepared another figure and uploaded just as “Fig 1”.

C9. We note that Figures 2, 3, 4 & 5 in your submission contain [map/satellite] images which may be copyrighted. All PLOS content is published under the Creative Commons Attribution License (CC BY 4.0), which means that the manuscript, images, and Supporting Information files will be freely available online, and any third party is permitted to access, download, copy, distribute, and use these materials in any way, even commercially, with proper attribution. For these reasons, we cannot publish previously copyrighted maps or satellite images created using proprietary data, such as Google software (Google Maps, Street View, and Earth). For more information, see our copyright guidelines: http://journals.plos.org/plosone/s/licenses-and-copyright.

R9. Figures 2, 3, 4, & 5 are removed from submission. 

C10. We suggest you thoroughly copyedit your manuscript for language usage, spelling, and grammar. If you do not know anyone who can help you do this, you may wish to consider employing a professional scientific editing service. 

R10. I confirm you that my manuscript was thoroughly copy edited for language usage, spelling, and grammar by my colleague. Please visit the tracked changes uploaded as “Revised manuscript with track changes”.

C11. The name of the colleague or the details of the professional service that edited your manuscript.

R11. The name of my colleague who edited my manuscript is called “Assefa Andargie”.

C12. A copy of your manuscript showing your changes by either highlighting them or using track changes (uploaded as a *supporting information* file). A clean copy of the edited manuscript (uploaded as the new *manuscript* file)

R12. Performed well. 

Reviewers’ Comments 

C13. The Authors are trying to gauge the road traffic noise pollution by evaluating the measured noise levels against the Ethiopian Standard and WHO Standard. However, the parameters of the criteria have not been indicated clearly. For instance, how do Ethiopian Standard and WHO Standard assess the noise levels? Is that Leq, L10, or L90? How long is the duration of the measurements? 

Noise is fluctuating energy, and thus the measurement duration plays an essential role for those statistical parameters. If the statistical parameter used to express the sound levels have not been indicated clearly, the measurement results are meaningless.

R13. Thank you very much for your information. Ethiopian standard to assess noise levels is Equivalent noise levels (Leq). The sampling time for each measurement was 30 minutes. Based on your comments, I clearly stated in the methods section. 

C14. Also, I suggest showing the measured noise levels tabulated in Table 1 to be illustrated in chart format for ease of reference.

R14. Thanks a lot. I changed to chart and named as “Fig 1”. However, I uploaded the earlier table for details information just as “supporting file”.

C15. Overall, this paper presented the study on noise pollution in Dessie City by considering noise measurement at health facilities area, residential areas and commercial areas. However, I did not find any contribution or new ideas presented in this paper. It is mainly reporting the outcomes of the measurement result without providing any new ideas or mitigation procedure to control the noise pollution.

R15. Thanks. Road traffic noise and its effect on the residents is widely studied especially in developed world. Thus this study may not provide new things here. However, the level of road traffic noise pollution is not well known in cities of developing countries, including Ethiopia as well as not compared with the respective standards/permissible levels. Filling this gap and make recommendations for concerned bodies is the objective of this study. Based on this understanding and your suggestion, I have revised the manuscript at discussion section. 

C16. The measurement result does not compared with the established standard (ISO 1996-1, etc.). The authors should refers these established standard in the discussion.

R16. Thank you so much. Revised. 

C17. The presented measurement result should also describe the noise in term of Lmax, L10, and LAeq as these are the standard value used in any noise analysis.

R17. Really thank you very much. I have revised in methods, results and discussion sections 

C18. It is always good if the authors can provide a new idea or new mitigation procedure in order to reduce noise pollution in urban areas.

R18. The authors revised and included mitigation procedures in the discussion section. 

C19. The authors mentioned about "professional judgement". I think the authors should described in details what type of judgement were considered here.

R19. It is described well. Through I consider WHO community-noise guideline, the author decided to select some sampling points as per the knowledge/experience of the author and current situations of the field sites.

C20. In my opinion, this paper required a lots of improvement in term of technical discussion, standard comparison and contribution in order to meet the requirement to publish in this Journal.

R20. Thanks again. I tried to incorporate all your comments. Kindly request you to visit the tracked changes document.

---

## [Decision Letter · Decision Letter 1]

4 May 2022

PONE-D-21-28974R1Spatial distribution of health-risky road traffic noise pollution in Dessie City, North East EthiopiaPLOS ONE

Dear Dr. Bogale,

Thank you for submitting your manuscript to PLOS ONE. After careful consideration, we feel that it has merit but does not fully meet PLOS ONE’s publication criteria as it currently stands. Therefore, we invite you to submit a revised version of the manuscript that addresses the points raised during the review process.

Please see below the comments and suggested MINOR revisions made by the individual(s) who reviewed your manuscript.  If provided, the referee's report(s) indicate the revisions that need to be made before it can be accepted for publication.

We look forward to receiving your revised manuscript.

Kind regards,

Ricardo Santos

Academic Editor

PLOS ONE

Journal Requirements:

Reviewers' comments:

Reviewer's Responses to Questions

**Comments to the Author**

1. If the authors have adequately addressed your comments raised in a previous round of review and you feel that this manuscript is now acceptable for publication, you may indicate that here to bypass the “Comments to the Author” section, enter your conflict of interest statement in the “Confidential to Editor” section, and submit your "Accept" recommendation.

Reviewer #1: All comments have been addressed

Reviewer #2: All comments have been addressed

2. Is the manuscript technically sound, and do the data support the conclusions?

Reviewer #1: Yes

Reviewer #2: Partly

3. Has the statistical analysis been performed appropriately and rigorously? 

Reviewer #1: Yes

Reviewer #2: Yes

4. Have the authors made all data underlying the findings in their manuscript fully available?

Reviewer #1: Yes

Reviewer #2: No

5. Is the manuscript presented in an intelligible fashion and written in standard English?

Reviewer #1: Yes

Reviewer #2: Yes

6. Review Comments to the Author

Reviewer #1: All previous comments have been addressed by the authors. Therefore, the manuscript is acceptable.

Reviewer #2: Thank you very much for addressing all my previous comment. However, I think the author should put more data into this paper. I described in details as below:

1) In your data analysis and interpretation, you already mentioned about LAeq, L10 and Lmax, please provide all the measurement data according to these parameter (seven days).

2) In discussion part, the author mentioned about WHO standard to be at (75dB and 45dB). It will be good if the author can provide the standard table of noise level that it referred to.

3) In discussion part, line 231 and 232, the author mentioned the result above standard LAeq level which are 65dB and 70dB. The result referred here is from Figure 1 and Table 1? If so, it is appropriate for the average value per day compared to LAeq standard value?

4) My recommendation is it will be good for the author to be able to present the plotting for LAeq, etc. for comparing the value with the standard values.

7. PLOS authors have the option to publish the peer review history of their article (what does this mean?). If published, this will include your full peer review and any attached files.

Reviewer #1: No

Reviewer #2: No

---

## [Author Response · Author response to Decision Letter 1]

31 May 2022

Dear Ricardo Santos,

We appreciate your decision to my manuscript and also thank you all the reviewers for your valuable MINOR comments. As per the instructions given by you, we made some corrections to the manuscript and provided responses (rebuttal letter) to each comments given by you and reviewer 2 as follow.

Authors response to Editor and Reviewers (note: - Comments=C1-11, Response=R1-11)

Manuscript ID: PONE-D-21-28974

Manuscript Title: Spatial distribution of health-risky road traffic noise pollution in Dessie City, North East Ethiopia

Author: Getahun Gebre Bogale 

Editor’s comments /queries 

C1. A rebuttal letter that responds to each point raised by the academic editor and reviewer(s). You should upload this letter as a separate file labeled 'Response to Reviewers'.

R1. Uploaded as 'Response to Reviewers 2'

C2. A marked-up copy of your manuscript that highlights changes made to the original version. You should upload this as a separate file labeled 'Revised Manuscript with Track Changes'.

R2. Uploaded as 'Revised Manuscript with Track Changes 2'

C3. An unmarked version of your revised paper without tracked changes. You should upload this as a separate file labeled 'Manuscript'.

R3. Uploaded as 'Manuscript 2'

C4. R4. No changes in financial disclosure.

Journal Requirements:

C5. Please review your reference list to ensure that it is complete and correct. If you have cited papers that have been retracted, please include the rationale for doing so in the manuscript text, or remove these references and replace them with relevant current references. Any changes to the reference list should be mentioned in the rebuttal letter that accompanies your revised manuscript. If you need to cite a retracted article, indicate the article’s retracted status in the References list and also include a citation and full reference for the retraction notice.

R5. The reference lists are complete and correct

Review Comments to the Author

Reviewer #1:

C6. All previous comments have been addressed by the authors. Therefore, the manuscript is acceptable.

R6. Thank you so much. 

Reviewer #2: Thank you very much for addressing all my previous comment. However, I think the author should put more data into this paper. I described in details as below:

C7. 1) In your data analysis and interpretation, you already mentioned about LAeq, L10 and Lmax, please provide all the measurement data according to these parameter (seven days).

R7. Thank you very much for your valuable comments. Since the main aim of this study is to show the distribution of health-risky road traffic noise pollution descriptively, we argued to present/interpret in such a way just to make our findings be a readers-friendly. We agreed that use of these technical parameters in detail may not be as such feasible for any lay readers. Of course, measurements are interpreted / compared according to LAeq, L10 and Lmax at discussion section. Few points are also incorporated to this version of revision. 

C8. 2) In discussion part, the author mentioned about WHO standard to be at (75dB and 45dB). It will be good if the author can provide the standard table of noise level that it referred to.

R8. Thanks a lot. The standard table of noise level was put at results section as “Table 1”, but it’s rephrased at discussion section. The guideline values for community noise is also indicated in the reference # 23.

C9. 3) In discussion part, line 231 and 232, the author mentioned the result above standard LAeq level which are 65dB and 70dB. The result referred here is from Figure 1 and Table 1? If so, it is appropriate for the average value per day compared to LAeq standard value?

R9. Thank you so much. Take corrections as ‘the results were compared to the standard Table 1 not from Figure 1’

C10. 4) My recommendation is it will be good for the author to be able to present the plotting for LAeq, etc. for comparing the value with the standard values.

R10. Thanks. See R7 above 

C11. While revising your submission, please upload your figure files to the Preflight Analysis and Conversion Engine (PACE) digital diagnostic tool, https://pacev2.apexcovantage.com/. PACE helps ensure that figures meet PLOS requirements. To use PACE, you must first register as a user. Registration is free. Then, login and navigate to the UPLOAD tab, where you will find detailed instructions on how to use the tool. If you encounter any issues or have any questions when using PACE, please email PLOS at figures@plos.org. Please note that Supporting Information files do not need this step.

R11. We confirm that Fig 1 meets PLOS requirements.

---

## [Editor Report · Decision Letter 2]

14 Jun 2022

Spatial distribution of health-risky road traffic noise pollution in Dessie City, North East Ethiopia

PONE-D-21-28974R2

Dear Dr. Bogale,

We’re pleased to inform you that your manuscript has been judged scientifically suitable for publication and will be formally accepted for publication once it meets all outstanding technical requirements.

Kind regards,

Ricardo Santos

Academic Editor

PLOS ONE
---

## [Editor Report · Acceptance letter]

6 Jul 2022

PONE-D-21-28974R2 

Spatial distribution of health-risky road traffic noise pollution in Dessie City, North East Ethiopia 

Dear Dr. Bogale:

I'm pleased to inform you that your manuscript has been deemed suitable for publication in PLOS ONE. Congratulations! Your manuscript is now with our production department. 

Kind regards, 

on behalf of

Dr. Ricardo Santos 

Academic Editor

PLOS ONE